# Assessing public support for extending smoke-free policies beyond enclosed public places and workplaces: protocol for a systematic review and meta-analysis

Nienke W Boderie ![ORCID] ,[1] Famke JM Mölenberg ![ORCID] ,[1] Aziz Sheikh,[2,3] Wichor M Bramer ![ORCID] ,[4] Alex Burdorf ![ORCID] ,[1] Frank J van Lenthe,[1] Jasper V Been[1,5]

For numbered affiliations see end of article.

**Correspondence to**
Dr Jasper V Been;
j.been@erasmusmc.nl

## ABSTRACT:

**Introduction** Smoke-free enclosed public environments are effective in reducing exposure to secondhand smoke and yield major public health benefits. Building on this, many countries are now implementing smoke-free policies regulating smoking beyond enclosed public places and workplaces. In order to successfully implement such 'novel smoke-free policies', public support is essential. We aim to provide the first comprehensive systematic review and meta-analysis assessing levels and determinants of public support for novel smoke-free policies.

**Methods and analysis** The primary objective of this review is to summarise the level of public support for novel smoke-free policies. Eight online databases (Embase.com, Medline ALL Ovid, Web of Science Core Collection, WHO Library Database, Latin American and Caribbean Health Sciences Literature, Scientific Online Library Online, PsychINFO and Google Scholar) will be searched from 1 January 2004 by two independent researchers with no language restrictions. The initial search was performed on 15 April 2020 and will be updated prior to finalisation of the report. Studies are eligible if assessing support for novel smoke-free policies in the general population (age ≥16 years) and have a sample size of n≥400. Studies funded by the tobacco industry or evaluating support among groups with vested interest are excluded. The primary outcome is proportion of public support for smoke-free policies, subdivided according to the spaces covered: (1) indoor private spaces (eg, cars) (2) indoor semiprivate spaces (eg, multi-unit housing) (3) outdoor (semi)private spaces (eg, courtyards) (4) non-hospitality outdoor public spaces (eg, parks, hospital grounds, playgrounds) and (5) hospitality outdoor public spaces (eg, restaurant terraces). The secondary objective is to identify determinants associated with public support on three levels: (1) within-study determinants (eg, smoking status) (2) between-study determinants (eg, survey year) and (3) context-specific determinants (eg, social norms). Risk of bias will be assessed using the Mixed Methods Appraisal Tool and a sensitivity analysis will be performed excluding studies at high risk of bias.

**Ethics and dissemination** No formal ethical approval is required. Findings will be disseminated to academics, policymakers and the general public.

## Strengths and limitations of this study

► This systematic review is unique in providing a structured overview of levels of public support for 'novel smoke-free policies' (ie, smoke-free policies that go beyond regulating smoking in enclosed public places and workplaces).

► Within-study and between-study determinants associated with public support will be assessed, and thematic synthesis will be used to identify context-specific determinants.

► The protocol presented has been designed in line with the Preferred Reporting Items for Systematic Reviews and Meta-Analysis Protocols.

► The generalisability and value of this systematic review depends on the availability and quality of the data.

## INTRODUCTION

Secondhand smoke (SHS) exposure is related to 1.2 million deaths per year.[1] Smoke-free environments have proven to be effective in reducing exposure to SHS and have major public health benefits.[2] Previous systematic reviews reported consistent evidence for improved cardiovascular health and reduced smoking-related mortality, as well as reductions in preterm birth, severe asthma exacerbations and respiratory tract infections in children, following implementation of smoke-free legislation in indoor public places and workplaces.[3–5] It has been shown that outdoor areas contribute significantly to SHS exposure, therefore the implementation of smoke-free policies in open spaces has the potential to reduce the associated burden of disease.[6 7]

In 2004, Ireland was the first country in the world to implement comprehensive smoke-free legislation covering enclosed workplaces and public places, and many more countries followed its example.[8] An increasing number

of jurisdictions is now implementing, or considering implementing, additional smoke-free policies that go beyond regulating smoking in enclosed public places and target private and outdoor spaces, henceforth referred to as 'novel smoke-free policies'. Novel smoke-free policies are implemented in an attempt to further improve population health via reducing SHS exposure. For example, several countries have implemented laws requiring private cars carrying children be smoke-free,[9–11] smoke-free hospital campuses have been implemented countrywide in Spain and Ireland,[12 13] the city of New York banned smoking in all public parks, pedestrian plazas and at all beaches,[14] and the US Department of Public Housing and Urban Development requires all public housing units to be smoke-free, both within resident units and in public areas.[15] Public support is essential in democracies in order for policymakers to consider implementing such novel smoke-free policies and to increase the likelihood of successful implementation,[16] and accordingly the WHO stated that 'Involving civil society is central to achieving effective legislation'.[5]

However, public support may vary over time, as well as by population subgroups. For example, women and non-smokers tend to be more in favour of smoke-free legislation than men and current smokers.[17] Several studies showed that public support for smoke-free policies increased after successful implementation and particularly so among smokers.[18–20] Furthermore, public support for smoke-free policies was higher when policies covered spaces that were frequently visited by those more vulnerable to the adverse health effects of SHS.[21] For example, in the USA and Canada public support for smoke-free playgrounds (89%–91%) was substantially higher than for smoke-free outdoor workplaces (12%–46%) and sidewalks (31%–49%).[17] Context-specific determinants may also contribute to differences in public support across settings. Aspects that enhanced successful adoption of smoke-free zones at outdoor school grounds at secondary schools included communication about the policy, collaboration between and within stakeholders, social norms and evidence about the effectiveness of smoke-free zones.[22]

A structured overview of the levels and determinants of public support for smoke-free policies beyond enclosed public places and workplaces across various settings is currently lacking. Having these insights may guide policymakers with the implementation of policies that receive the highest levels of support, and may help in defining additional strategies that are needed to increase public support in the population. To address this gap in the literature, our primary objective is to summarise the level of public support across the globe for novel smoke-free policies and to evaluate if public support changed following implementation of the novel smoke-free policies across various settings. To do so, a systematic review and meta-analysis will be conducted. The secondary objective is to identify determinants associated with public support at the following three levels: (1) within-study determinants

(eg, age, smoking status, parental status), (2) between-study determinants (eg, income level of the country, whether smoke-free legislation in enclosed public places and workplaces was already in place) and (3) context-specific determinants (eg, setting, framing, enforcement of smoke-free policies).

## METHODS AND ANALYSIS
We used the Preferred Reporting Items for Systematic Reviews and Meta-Analysis Protocols guidelines to facilitate development of this protocol, see online supplemental appendix 1.

In this review, we will use the term 'traditional smoke-free legislation' to refer to smoke-free legislation covering enclosed public places and workplaces (ie, compliant with Article 8 (2) of the Framework Convention on Tobacco Control) and the term 'novel smoke-free policies' to refer to policies and legislation regulating smoking in any other places, such as (semi)private places and (partially) outdoor spaces, whether public or (semi)private.[23] Policies are used in the broadest sense and are not necessarily enacted via formal legislation as this will allow us to evaluate less formal local smoke-free initiatives (eg, self-regulation by the hospitality sector or local hospitals) as well as formal legislation.

### Eligibility criteria
We will include articles published in scientific journals as well as 'grey literature' evaluating public support for novel smoke-free policies covering (semi)private places and (partially) outdoor spaces, whether public or (semi)private. Grey literature includes policy documents and reports that are published non-commercially and/or are not indexed by major scientific literature databases. Cohort studies and (repeated) cross-sectional studies will be included and no language restrictions are applied. Qualitative studies will be excluded. We will seek translation for reports in foreign languages to assess eligibility. Studies for which only an abstract is available will not be included since risk of bias for these studies cannot be adequately assessed.

Eligibility of the studies will be assessed using the following criteria:
1. Studies will be eligible if support for one or more novel smoke-free policies is evaluated. We will include studies assessing support for novel smoke-free policies that are already in place as well as those assessing support for upcoming or theoretical implementation of such policies. Policies at any level are eligible, such as city-level, state-level and country-level. Studies will be excluded if solely evaluating traditional smoke-free legislation.[23]
2. Studies will be eligible if they assessed public support for smoke-free policies in the population aged 16 years or above who represent the majority of a population primarily affected by the policy (eg, support for a country-wide measure is evaluated in a representative sample of the country, while support for a policy at a

local campus is assessed among students and staff of that specific campus), or in any of the predefined population subgroups (see *within-study determinants of public support* below). We set this age criterion to include the part of the population that is entitled to vote in most democracies, and as such may be regarded to be of particular interest to politicians and policymakers. Any study reporting (sub)populations in which at least 50% fits this age criterion will also be included.

3. Our primary objective is to summarise the level of public support for novel smoke-free policies in the general population, therefore we will only include studies of which we can be confident that the reported support in the study sample reflects the levels of support that would be found if the entire population was surveyed. This is operationalised by only including studies that can ensure a 5% margin of error. Following sample size calculations for surveys,[24 25] a minimum sample size of 400 is required. A similar criterion was used by an earlier review assessing public support for outdoor smoke-free areas.[17]

4. Studies will be included when published from 1 January 2004 onwards. This pragmatic cut-off chosen as the first national traditional smoke-free law covering indoor public places and workplaces was introduced in Ireland in 2004. Hence, assessments of public support for novel smoke-free policies are unlikely to have preceded 2004, and are unlikely to be relevant for current everyday practice if they have.

5. Studies will be excluded if solely evaluating support among specific subgroups not representing the majority of the population primarily affected by the policy, policymakers or groups with clearly vested interest, for example, opinion of tobacco industry groups.

6. Studies will be excluded when funded or supported by the tobacco industry, as the tobacco industry is known to 'produce, sponsor and disseminate misleading research and information, lacking sound scientific methods'.[26]

7. Studies will be excluded if solely evaluating support for tobacco-related subgroups, for example, e-cigarettes or heatless tobacco products.

## Information sources
The following electronic databases will be searched for eligible studies: (1) Embase.com, (2) Medline ALL Ovid, (3) Web of Science Core Collection, (4) WHO Library Database, (5) Latin American and Caribbean Health Sciences Literature, (6) Scientific Online Library Online, (7) PsychINFO and (8) Google Scholar.

## Search strategy
The specific search strategies per database have been created in close collaboration with a bibliographical expert of the Erasmus MC with expertise in systematic reviewing (WMB; see online supplemental appendix 1). Search terms include three parts: (1) terms to identify smoke-free policies; (2) terms to identify measures of public support as the outcome; and (3) terms that exclude letters to the editors, notes and editorials.

We will complement our search by screening reference lists of reviews related to the topic and of included studies and their citations through Scopus, following Bramer.[27] We will update our search to add the most recent reports just before submitting our final review report for publication.

## Study records
### Data management
All records identified by the search strategy will be extracted into an EndNote Library, and we will de-duplicate using this software following the procedure outlined by Bramer *et al*.[28] If any duplicates remain, those will be manually excluded. At this stage, duplicates will be identified based on overlapping author names and titles. The total number of detected duplicates will be noted in the final report.

### Selection process
After removing duplicates, titles and abstracts of records identified during the literature search will be screened independently for inclusion by two reviewers. After initial selection based on screening of titles and abstracts, full-text articles will be screened for eligibility according to the inclusion and exclusion criteria by two reviewers, and discrepancies will be resolved after discussion with a third reviewer. Remaining duplicates based on populations, sample size and reported outcomes will be identified based on full text. The reviewers will not be blinded to information about the articles (eg, authors' names and affiliations) at any stage.

### Data collection process
Two reviewers will independently extract relevant data from all included studies according to a customised data extraction form developed a priori that was piloted using four eligible studies. On completion the reviewers will compare their results and any discrepancies will again be resolved after discussion with a third reviewer. If any relevant data are missing, the corresponding authors will be contacted.

### Data items
Customised data extraction forms will be used to extract relevant information from the eligible studies, which will include the following items:
1. First author's name and affiliation.
2. Publication year.
3. Type of publication.
4. Access information (DOI or URL).
5. Study design.
6. Location of the study (eg, country, region).
7. Description of the policy (eg, places covered, whether or not the intervention is implemented, national or regional/local implementation).
8. For studies assessing support for policies already implemented:

a. Date of implementation.
b. Level of implementation (eg, government, municipality).
c. Level of enforcement (eg, voluntary, warnings, fines).
9. Observational period.
10. Selection of participants (eg, eligibility criteria, sampling methods).
11. Number of participants.
12. Data source (eg, national survey, study recruited participants).
13. Method of data collection.
14. Definition of public support.
15. Statistical analyses (if applicable).
16. Number and percentages of missing values and non-response (if applicable).
17. Techniques for handling missing values and non-response.
18. Characteristics of the study population (eg, age, gender, smoking status).
19. Levels of public support (estimate, 95% CIs).
20. Determinants of public support (see the Data synthesis section for more detail).
    a. Within-study determinants.
    b. Between-study determinants.
    c. Context-specific determinants.
21. Any conflict of interest reported by the authors.
22. Funding source(s).

Data will be complemented with the World Bank Country Classification by income, based on Gross National Income per capita.[29] Furthermore, we will seek information regarding whether at the time of the study traditional smoke-free regulation was already implemented in enclosed public areas and workspaces according to the WHO.[23]

## Outcomes and prioritisation

Data will be extracted for each estimate of public support by the spaces that they cover (eg, playgrounds, private cars, multi-unit housing). If weighted and unweighted estimates are presented, we will extract estimates that are weighted to most adequately reflect the general population. If multiple estimates are presented that relate to public support, we will extract the estimate that covers the most general spaces. For example, we will prioritise 'it should be illegal to smoke in all playgrounds' above 'it should be illegal to smoke in this specific playground'. If public support is asked in general and specifically related to children, we will extract both estimates. For example, we will extract 'it should be illegal to smoke in private cars' and 'it should be illegal to smoke in private cars when minors are present'.

## Risk of bias assessment

We will assess risk of bias for each study using the Mixed Methods Appraisal Tool (MMAT) for descriptive studies. The MMAT 2018 version was developed based on criteria from 18 existing critical appraisal tools and input from over 50 international experts. The following five elements will be assessed: relevance of the sampling strategy, representativeness of the target population, appropriateness of the outcome measurements, risk of non-response bias and appropriateness of the statistical techniques. Each of the elements will be categorised by using the answer categories *yes*, *no* or *can't tell*, following MMAT criteria. Results of the risk-of-bias analysis will be presented in tables.[30]

## Data synthesis

Obtaining comparable data is essential to facilitate meta-analysis, thus homogenisation of the outcome data is needed. Public support will be analysed as proportional data, that is, proportion of the population supporting a particular smoke-free policy. When results are fairly normally distributed the raw proportions will be analysed, if not logit transformations will be applied.[31] The outcome estimates will be reversed if studies report on the proportion not in favour of the smoke-free policies. Often Likert scale-type questions are used to assess support; if studies report percentages per answer option instead of total support, the answer categories above neutral (ie, indicating a positive response) will be combined.

To allow meta-analyses, SEs are needed. If SEs are not presented, they will be calculated using the following formula[32]:

$$fraction = \frac{nr\ of\ people\ supporting\ policy}{sample\ size}$$

$$SE = \sqrt{\frac{fraction(1-fraction)}{sample\ size}}$$

Two reviewers will independently assess whether measures of public support and smoke-free policies under investigation are sufficiently comparable across the selected studies to allow meta-analysis. If needed, they will convert the units of measurement in a way that is consistent across studies. In case of disagreement, a third reviewer will decide which measures to use.

Prior to undertaking meta-analyses, we will subdivide policies by spaces that they cover according to the following division: (1) indoor private spaces (eg, cars), (2) indoor semiprivate spaces (eg, multi-unit housing), (3) outdoor (semi)private spaces (eg, courtyard, psychiatric hospital), (4) non-hospitality outdoor public spaces (eg, parks, streets, beaches, hospital grounds, playgrounds) and (5) hospitality outdoor public spaces (eg, restaurant terraces). Separate meta-analyses will be conducted to assess public support for smoke-free policies according to these categories. If multiple estimates of public support are presented that cover similar spaces according to our categorisation, we will calculate the average public support across these spaces for use in meta-analyses. Thus, if studies present separate estimates of public support for playgrounds, parks and beaches (all belonging to the category 'non-hospitality outdoor public spaces'), the average of the three will be used. In case of overlapping study samples, we will include the study or effect estimation that: (1) uses national surveys and therefore is likely

to be representative of the general population, (2) has the lowest risk of bias or (3) incorporates the largest sample size, following this hierarchy. We aim to summarise the most up-to-date status of public support for novel smoke-free policies; thus, we will include the most recent estimation if studies presented multiple estimates over time. In secondary analyses, we will evaluate whether public support changed following the actual introduction of the smoke-free policy under study, if the data allow.

One of the assumptions in meta-analysis is that effect sizes are independent, that is, the effect size of one study does not imply the direction or magnitude of the effect size in another study.[33] Multiple estimates of public support in a specific country may violate the independence assumption, therefore we will conduct a three-level meta-analysis.[34] A three-level meta-analysis is an extended version of a random-effects meta-analysis and includes sampling variation at the first level, within-country heterogeneity at the second level and between-country heterogeneity at the third level. The analytical model is as follows:

$$(1)\ \hat{\theta}_{ij} = \beta_0 + \zeta_{(2)ij} + \zeta_{(3)j} + \epsilon_{ij}.$$

Where $\hat{\theta}_{ij}$ is the estimation of the true effect size for public support, $\beta_0$ the average population effect, $\zeta_{(2)ij}$ is the within-country variance, $\zeta_{(3)j}$ is the between-country variance and $\epsilon_{ij}$ the sampling variance.[34] In each model, heterogeneity will be quantified by the $I^2$ statistic per level. We intend to use R V.3.6.5 (R Foundation for Statistical Computing, 2020) using the packages meta and metaphor for all analyses.[35 36] A second analysis will be performed on all studies presenting the change in public support following implementation of the actual policy under study. If relevant analyses will be performed twice, once including estimates for public support in general and once for public support specifically related to children, for example, public support for smoking bans in cars and public support for smoking bans in cars if children are present (see the Outcomes and prioritisation section).

The secondary objective is to identify and quantify determinants that are associated with public support. The determinants of public support will be evaluated at three levels:

1. Quantify within-study determinants of public support. Public support may differ between population subgroups. Therefore, we will conduct subgroup analyses according to:
► Gender (men vs women).
► Smoking status (current smokers vs former smokers vs non-smokers, and/or current smokers vs non-smokers (including former smokers), depending on data availability).
► Parental status (yes vs no, depending on data availability).
► Age group (younger vs older, categorisation depending on data availability).

Public support will be pooled per subgroup for each of the five spaces categories using random-effects three-level meta-analysis.[37]

2. Quantify between-study determinants of public support. Various study-specific elements may influence public support. Random-effects linear meta-regression analysis will be performed to assess between-study determinants of public support according to the various spaces that the policies cover. In these analyses support in favour of smoke-free policies is used as dependent variable and the following variables per study are used as independent variables:
► Calendar year in which the survey was conducted (continuous).
► Whether public support was assessed as yes–no or on a Likert scale (binary).
► Income level of the country, according to the World Bank classification (binary: high-income vs low-income and middle-income countries).
► Whether or not traditional smoke-free legislation covering enclosed public places and workplaces was in place (categorical: none, partial or comprehensive according to the WHO classification).[23]

3. Identify context-specific determinants of public support.

Context-specific determinants of public support will be identified using thematic synthesis. We will follow the method outlined by Thomas and Harden consisting of three steps[38]: (1) coding text, (2) developing descriptive themes and (3) generating analytical themes. The full text of each study will be extracted and uploaded into NVivo V.12 (NVivo Qualitative data analysis software V.10: QSR International, 2012). As studies may provide information outside the scope of this review, coding will be limited to sentences describing details that relate to determinants of public support for smoke-free policies. In order to ensure a consistent coding methodology, three eligible articles will be coded independently by two reviewers and then compared until consensus on the themes has been reached. The remaining articles will be coded independently by two reviewers. After every five articles, coding will be compared to ensure consistency. A priori four core domains have been identified: (1) beliefs and scientific evidence about effectiveness, (2) social norms, (3) communication and implementation strategies, and (4) collaboration between stakeholders.

## Sensitivity analysis
Study findings may vary according to the risk-of-bias level of the individual studies. As a sensitivity analysis, we will exclude studies that scored *no* or *can't tell* on at least one domain following the MMAT criteria. This criterion is based on MMAT evaluations in previous literature.[39]

## Ethics and dissemination
No primary data collection will be undertaken; therefore, no formal ethical assessment and informed consent are required. Findings will be summarised in a single manuscript and will be disseminated through scientific literature.

## Timeline
Start date: 15 April 2020.
  Finishing date: 31 May 2021.

**Author affiliations**
¹Department of Public Health, Erasmus MC, University Medical Centre Rotterdam, Rotterdam, The Netherlands
²Centre of Medical Informatics, Usher Institute, University of Edinburgh, Edinburgh, UK
³Division of General Internal Medicine and Primary Care, Brigham and Women's Hospital/Harvard Medical School, Boston, Massachusetts, USA
⁴Medical Library, Erasmus MC, University Medical Centre Rotterdam, Rotterdam, The Netherlands
⁵Division of Neonatology, Department of Paediatrics, Erasmus MC - Sophia Children's Hospital, University Medical Centre Rotterdam, Rotterdam, Netherlands

**Contributors** JVB conceptualised the study and secured funding. All authors contributed to the design of the protocol. WMB developed the search strategy. NWB and FM wrote the first draft and revised subsequent drafts. JVB supervised the writing. AS, WMB, AB and FJVL contributed to the writing, and all authors have read and approved the final manuscript.

**Funding** This work was funded by a joint collaboration between the Lung Foundation Netherlands, Dutch Cancer Society, Dutch Heart Foundation, Dutch Diabetes Research Foundation and the Netherlands Thrombosis Foundation (grant number 2.1.19.010). AS is supported by Health Data Research UK.

**Competing interests** None declared.

**Patient and public involvement statement** Patients and/or the public were not involved in the design, or conduct, or reporting, or dissemination plans of this research.

**Patient consent for publication** Not required.

**Provenance and peer review** Not commissioned; externally peer reviewed.

**ORCID iDs**
Nienke W Boderie http://orcid.org/0000-0002-1600-380X
Famke JM Mölenberg http://orcid.org/0000-0002-5305-9730
Wichor M Bramer http://orcid.org/0000-0003-2681-9180
Alex Burdorf http://orcid.org/0000-0003-3129-2862

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
