## [Reviewer comments · BMJ Open]

ARTICLE DETAILS

TITLE (PROVISIONAL)	Assessing public support for extending smoke-free policies beyond enclosed public places and workplaces: protocol for a systematic review and meta-analysis
AUTHORS	Boderie, Nienke; Mölenberg, Famke; Sheikh, Aziz; Bramer, Wichor; Burdorf, Alex; Van Lenthe, Frank; Been, Jasper

VERSION 1 – REVIEW

REVIEWER	Kate Frazer University College Dublin Ireland
REVIEW RETURNED	19-May-2020

GENERAL COMMENTS	Congratulations on developing a systematic review protocol and considering the extension of smoke-free policies. This is a well thought out protocol. I wish you well and look forward to seeing the results.
---

REVIEWER	Esteve Fernández Catalan Institute of Oncology / University of Barcelona
REVIEW RETURNED	03-Jun-2020

GENERAL COMMENTS	The manuscript presents the protocol for a systematic review and meta-analysis on public support for extending smoke-free policies. The protocol is well written and considers the full process for systematic review and meta-analysis. Some comments follow: 1. Introduction. As part of the justification the authors should mention that exposure to secondhand smoke (SHS) in open and semi-open spaces have been objectively assessed. Two previous reviews addressed it, and also elaborated about the possibility of extending smoke-free legislation to outdoor places (please see Sureda et al. Environ Health Perspect. 2013;121(7):766-73 and Licht et al. Tob Control. 2013;22(3):172-9).2. Introduction. In the second paragraph, there is an example of smoke-free legislation beyond enclosed public places that, in fact, refers to enclosed private places (smoking in cars). The authors could include examples of current smoke-free legislation covering also outdoor public places: Spain and Romania, for example, have smoke-free legislation regulating smoking in outdoor terraces of restaurants, pubs/bars and discos/nightclubs, and also in outdoor campuses of schools and hospitals enacted in 2011 and 2016, respectively.
---

	3. Methods. The authors should provide a more complete definition of what will be consider a "policy". Are "interventions" (at different levels, local, specific collectives such as young, workers...) considered "policies"? What do they consider by "not necessarily enacted via formal legislation"? Self-regulation by the hospitality sector would be considered as policy? 4. Methods. Eligibility of studies. The authors should consider to list as a separate criteria the exclusion of tobacco industry funded studies, and add some rationale for it. 5. Methods: For the primary aim, which will be the model specification (linear, logistic, Poisson...)? 6. Methods: For the secondary aims, will the authors use any type of measure of association (prevalence ratio, odds ratio ?) and from which type of model will be derived?
--	--

REVIEWER	Satyanarayana Labani Previously at ICMR-NICPR, Noida , India
REVIEW RETURNED	07-Sep-2020

GENERAL COMMENTS	Comments to authors:  1. Abstract – Introduction (last line) and Abstract-Methods (1st line) are different. 2. Basis is given for n=400 for inclusion is not justifiable. Even lower than 400 sample size studies also be useful and cannot afford to omit them in meta-analysis. 3. Meta-analysis needs to be included in the objective as it is present in the title. 4. Exclusion of right at the beginning of studies related to conflict of interest. They can be assessed in the sensitivity analysis. 5. Study designs of the primary studies to considered is not stated in the protocol. 6. Why MMAT is chosen by other many tools frequently used are considered requires to be stated by authors. 7. This protocol is not under consideration by Prospero. There is another study by authors for less than 16years was found there. This could not be located. 8. Authors may add explanation of why they need a structured overview of the levels and determinants of public support for smoke-free policies beyond enclosed public places and workplaces across various settings. 9. Statistical software / package proposed to use is not stated in methods.
--

VERSION 1 – AUTHOR RESPONSE

Reviewer #1 to the authors:

Congratulations on developing a systematic review protocol and considering the extension of smoke-free policies. This is a well thought out protocol. I wish you well and look forward to seeing the results.

Response: We are grateful for your kind words and look forward to share the results with you in the near future.

Reviewer #2 to the authors:

The manuscript presents the protocol for a systematic review and meta-analysis on public support for extending smoke-free policies. The protocol is well written and considers the full process for systematic review and meta-analysis. Some comments follow:

1. Introduction. As part of the justification the authors should mention that exposure to secondhand smoke (SHS) in open and semi-open spaces have been objectively assessed. Two previous reviews addressed it, and also elaborated about the possibility of extending smoke-free legislation to outdoor places (please see Sureda et al. *Environ Health Perspect.* 2013;121(7):766-73 and Licht et al. *Tob Control.* 2013;22(3):172-9).

Response: Thank you for suggesting these two systematic reviews on SHS in outside areas. In addition to legislation in indoor public places and workplaces, we now also refer to the possible link between smoke-free policies in outdoor areas and health outcomes. In the introduction we refer to the reviews on SHS exposure in outside areas kindly suggested by the reviewer (p. 3, line 7-9):

“Previous systematic reviews reported consistent evidence for improved cardiovascular health and reduced smoking-related mortality, as well as reductions in preterm birth, severe asthma exacerbations and respiratory tract infections in children, following implementation of smoke-free legislation in indoor public places and workplaces.¹⁻³ It has been shown that outdoor areas contribute significantly to SHS exposure, and implementation of smoking-free policies in open spaces has the potential to reduce the associated burden of disease.^{4, 5}”

2. Introduction. In the second paragraph, there is an example of smoke-free legislation beyond enclosed public places that, in fact, refers to enclosed private places (smoking in cars). The authors could include examples of current smoke-free legislation covering also outdoor public places: Spain and Romania, for example, have smoke-free legislation regulating smoking in outdoor terraces of restaurants, pubs/bars and discos/nightclubs, and also in outdoor campuses of schools and hospitals enacted in 2011 and 2016, respectively.

Response: In addition to the references about smoke free private places we have now added additional examples (p. 3, line 12-16) regarding smoke free outdoor public places to stress that novel smoke free policies in our paper refer to both indoor (semi-) private places and outdoor public and (semi-) private places with smoking regulations. New text at p3. Line 12 to 16:

“An increasing number of jurisdictions is now implementing, or considering implementing, additional smoke-free policies that go beyond regulating smoking in enclosed public places and target private and outdoor spaces, henceforth referred to as ‘novel smoke-free policies’. Novel smoke-free policies are implemented in an attempt to further improve population health via reducing SHS exposure. For example, several countries have implemented laws requiring private cars carrying children be smoke-free,⁶⁻⁸ smoke-free hospital campuses have been implemented country-wide in Spain and Ireland,⁹ ,¹⁰ the city of New York banned smoking in all public parks, pedestrian plazas and at all beaches,¹¹ and the US Department of Public Housing and Urban Development requires all public housing units to be smoke-free, both within resident units and in public areas.¹² “

3. Methods. The authors should provide a more complete definition of what will be consider a "policy". Are "interventions" (at different levels, local, specific collectives such as young, workers...) considered "policies"? What do they consider by "not necessarily enacted via formal legislation"? Self-regulation by the hospitality sector would be considered as policy?

Response: We use the word policy in a broad sense, thus referring to any set of ideas or plans agreed upon by a group of people. By adding “not necessarily enacted via formal legislation” we allow less formal initiatives such as the example provided by the reviewer to be included. We have now extended our explanation in our revised paper at p. 4, line 21-23:

“Policies are considered in the broadest sense and are not necessarily enacted via formal legislation. This will allow us to evaluate less formal local smoke-free initiatives (e.g. self-regulation by the hospitality sector or by local hospitals) as well as formal legislation.”

4. Methods. Eligibility of studies. The authors should consider to list as a separate criteria the exclusion of tobacco industry funded studies, and add some rationale for it.

Response: Thank you for the suggestion. We strongly agree that tobacco industry funded studies should be excluded. To stress this importance we have added the exclusion criterion and rationale to p.5, line 33-35. The criterion is formulated as follows:

“Studies will be excluded when funded or supported by the tobacco industry, as the tobacco industry is known to “produce, sponsor and disseminate misleading research and information, lacking sound scientific methods” . “

5. Methods: For the primary aim, which will be the model specification (linear, logistic, Poisson...)?

Response: Our primary objective is to summarise the level of support for novel smoke-free policies, thus using the proportion of the population supporting a particular smoke-free policy in our analysis. When using proportional data it depends on the distribution of the data whether crude proportions can be analysed or whether logit transformations should be applied. This consideration and the following options has been added to our protocol (p.8, line 6-8):

“Public support will be analysed as proportional data, i.e. proportion of the population supporting a particular smoke-free policy. When results are at least reasonably normally distributed the crude proportions will be analysed, if not logit transformations will be applied.13”

6. Methods: For the secondary aims, will the authors use any type of measure of association (prevalence ratio, odds ratio ?) and from which type of model will be derived?

Response: Our secondary objective is to identify determinants that are associated with public support at three levels; within studies, between studies, and context specific. The within study determinants will be identified through subgroup analyses, for which random-effects meta-analysis of proportions will be performed. This will result in percentage of support (and corresponding 95%CI) per subgroup. See p.9, line 30-31: “Public support will be pooled per subgroup for each of the five spaces categories using random-effects three-level meta-analysis.14 ”

The between study determinants will be identified through random effects linear meta-regression, see p.9, line 33-35: “Various study-specific elements may influence public support. Random-effects linear meta-regression analysis will be performed to assess between-study determinants of public support according to the various spaces that the policies cover”. Results from these analyses will also be reported as difference in absolute percentages including 95%CI.

Reviewer #3 to the authors: Comments to authors:

1. Abstract – Introduction (last line) and Abstract-Methods (1st line) are different.

Response: Our aim is to provide a comprehensive systematic review and meta-analysis assessing levels and determinants of public support for novel smoke-free policies, which will be reached through two objectives: 1. summarising the level of public support for novel smoke-free policies, and 2. identifying determinants associated with public support. We have reformulated our primary and secondary aim as “objectives” throughout the document (p. 2, line 9, 18; p. 3, line 43, p.4 line 8 etc.)
2. Basis is given for n=400 for inclusion is not justifiable. Even lower than 400 sample size studies also be useful and cannot afford to omit them in meta-analysis.

Response: Our primary aim is to summarise the level of public support for novel smoke free policies in the general population. Therefore, we restrict our study inclusion to studies of which we can be confident that the reported support in the study sample reflects the levels of support that would be found if the entire population was surveyed. Cochran's formula is commonly used to determine appropriate sample sizes in survey research.¹⁵ Bartlett and colleagues described the procedures for determining sample sizes for categorical variables.¹⁶ Determining the sample sizes requires to set the alpha level and the margin of error.

In our study, we set the alpha level at 0.05, and the acceptable margin of error at 5%. For the proportion of public support, a 5% margin error would result in the researcher being confident that the proportion of respondents who were in favour of a specific policy was within $\pm 5\%$ of the proportion calculated from the sample size. Using this formula, based on a 5% error margin and alpha set at 0.05, a sample size of 384 responses was deemed necessary.

It might be true that for smaller populations or population subgroups, smaller sample sizes would also be sufficient. However, the main interest of our paper is to estimate the general support for smoke-free policies in the population, which we feel justifies our decision to exclude studies with a sample size below 400 respondents. A previous literature study on public support for smoke-free outdoor areas in the USA and Canada also used this number as a cut-off.¹⁷

We explained in more detail why excluding studies with a sample size below 400 respondents was deemed appropriate at p.5, line 13 to 19:

“Our primary objective is to summarise the level of public support for novel smoke free policies in the general population, therefore we will only include studies of which we can be confident that the reported support in the study sample reflects the levels of support that would be found if the entire population was surveyed. This is operationalized by only including studies that can ensure a 5% margin of error. Following sample size calculations for surveys,¹⁵ ¹⁶ a minimum sample size of 400 is required. This criterion is in keeping with an earlier review assessing public support for outdoor smoke-free areas.¹⁷”

3. Meta-analysis needs to be included in the objective as it is present in the title.

Response: We have now added meta-analysis to the study objective at p.2, line 7:

“We aim to provide the first comprehensive systematic review and meta-analysis assessing levels and determinants of public support for novel smoke-free policies.”

4. Exclusion of right at the beginning of studies related to conflict of interest. They can be assessed in the sensitivity analysis.

Response: In this study we aim to summarize the evidence about public support for novel smoke-free policies, that may inform policy makers considering these policies. We therefore strongly believe that studies with conflicts of interest should not be included. Although it could be interesting to see if differences in support are present among studies with and without conflict of interest, this is beyond the scope of our review. We aim to summarise unbiased estimates of support.

5. Study designs of the primary studies to considered is not stated in the protocol.

Response: We did not apply restrictions on study designs other than the exclusion of qualitative studies. However we do anticipate that mainly cohort studies and repeated cross-sectional studies will be included. We have added this to p. 4, line 31-33:

“Cohort studies and (repeated) cross-sectional studies will be included, qualitative studies will be excluded and no language restrictions are applied.”

6. Why MMAT is chosen by other many tools frequently used are considered requires to be stated by authors.

Response: We anticipate that mainly (repeated) cross-sectional studies and perhaps some cohort studies will be included. The MMAT is a validated tool to assess risk of bias in cross-sectional studies. The tool has been frequently updated, and the last version of 2019 was developed with the input with over 50 experts from various backgrounds and using information from 18 existing appraisal tools.18 MMAT version 2011 has been cited by over 400 studies, and the newest version of the MMAT 2018 has been cited over a 150 times. We now added some of this information to our protocol on p. 7, line 40-41:

“The MMAT 2018 version was developed based on criteria from 18 existing critical appraisal tools and using input from over 50 international experts”

7. This protocol is not under consideration by Prospero. There is another study by authors for less than 16 years was found there. This could not be located.

Response: We deliberately postponed PROSPERO registration of our protocol so as to enable feedback from peer-reviewers to be incorporated in our protocol. We have now registered our protocol and will provide the registration details in the revised manuscript.

8. Authors may add explanation of why they need a structured overview of the levels and determinants of public support for smoke-free policies beyond enclosed public places and workplaces across various settings.

Response: Having these insights will help guide policy makers with the implementation of policies that receive the highest levels of support, and may help in defining additional strategies that are needed to increase public support in the population. Also, information about the key determinants of public support may help guide efforts to increase support among subgroups of the population. As this is indeed important to explain in our protocol, we have added the previous explanation to p.3, line 40 to p.4 line 3:

“Having these insights may guide policy makers with the implementation of policies that receive the highest levels of support, and may help in defining additional strategies that are needed to increase public support in the population. To address this gap in the literature our primary objective is to summarise the level of public support across the globe for novel smoke-free policies and to evaluate if public support changed following implementation of the novel smoke-free policies across various settings. To do so a systematic review and meta-analysis will be conducted. ”

9. Statistical software / package proposed to use is not stated in methods.

Response: We intend to use R using the meta19 and metaphor20 package, this information is added to the protocol p.9,line 12 to 14:

“We intend to use R 3.6.5 (R Foundation for Statistical Computing, 2020) using the packages meta and metaphor for all analyses. 19 20

REFERENCES:

1. Faber T, Kumar A, Mackenbach JP, et al. Effect of tobacco control policies on perinatal and child health: a systematic review and meta-analysis. *Lancet Public Health* 2017;2(9):e420-e37.
2. Frazer K, Callinan JE, McHugh J, et al. Legislative smoking bans for reducing harms from secondhand smoke exposure, smoking prevalence and tobacco consumption. *Cochrane Database Syst Rev* 2016;2:CD005992.
3. World Health Organization. WHO Report on the Global Tobacco Epidemic, 2009: implementing smoke-free environments France: Geneva: World Health Organization, 2009.
4. Sureda X, Fernandez E, Lopez MJ, et al. Secondhand tobacco smoke exposure in open and semi-open settings: a systematic review. *Environ Health Perspect* 2013;121(7):766-73.

5. Licht AS, Hyland A, Travers MJ, et al. Secondhand smoke exposure levels in outdoor hospitality venues: a qualitative and quantitative review of the research literature. *Tob Control* 2013;22(3):172.
6. Thomson G, Weerasekera D, Wilson N. New Zealand smokers' attitudes to smokefree cars containing preschool children: Very high support across all sociodemographic groups. *New Zealand Med J* 2009;122(1300):84-86.
7. Patel M, Thai CL, Meng Y-Y, et al. Smoke-free car legislation and student exposure to smoking. *Pediatrics* 2018;141(Supplement 1):S40-S50.
8. Faber T, Mizani MA, Sheikh A, et al. Investigating the effect of England's smoke-free private vehicle regulation on changes in tobacco smoke exposure and respiratory disease in children: a quasi-experimental study. *Lancet Public Health* 2019;4(12):e607-e17.
9. Sureda X, Ballbè M, Martínez C, et al. Impact of tobacco control policies in hospitals: Evaluation of a national smoke-free campus ban in Spain. *Preventive Medicine Reports* 2014;1:56-61.
10. Fitzpatrick P, Gilroy I, Doherty K, et al. Implementation of a campus-wide Irish hospital smoking ban in 2009: prevalence and attitudinal trends among staff and patients in lead up. *Health Promot Int* 2009;24(3):211-22.
11. Johns M, Coady MH, Chan CA, et al. Evaluating New York City's smoke-free parks and beaches law: A critical multiplist approach to assessing behavioral impact. *American journal of community psychology* 2013;51(1-2):254-63.
12. Levy DE, Adams IF, Adamkiewicz G. Delivering on the Promise of Smoke-Free Public Housing. *American journal of public health* 2017;107(3):380-83.
13. Wang N. *How to Conduct a Meta-Analysis of Proportions in R: A Comprehensive Tutorial*, 2018.
14. Borenstein M, Higgins JPT. Meta-analysis and subgroups. *Prevention science* 2013;14(2):134-43.
15. Cochran WG. *Sampling Technique*. John Wiley and Son. New York 1977.
16. Bartlett JE, Kotrlik JW, Higgins CCH. Organizational research: Determining appropriate sample size in survey research appropriate sample size in survey research. *Inf Technol J* 2001;19(1):8.
17. Thomson G, Wilson N, Collins D, et al. Attitudes to smoke-free outdoor regulations in the USA and Canada: a review of 89 surveys. *Tob Control* 2016;25(5):506-16.
18. Hong QN, Pluye P, Fàbregues S, et al. *Mixed Methods Appraisal Tool (MMAT) version 2018: User guide*. Department of Family Medicine, McGill University 2018.
19. Schwarzer G. meta: An R package for meta-analysis. *R news* 2007;7(3):40-45.
20. Viechtbauer W. Conducting meta-analyses in R with the metafor package. *J Stat Softw* 2010;36(3):1-48.

VERSION 2 – REVIEW

REVIEWER	Esteve Fernández Catalan Institute of Oncology
REVIEW RETURNED	18-Nov-2020
GENERAL COMMENTS	The authors have done an excellent work and the manuscript has improved. No other comments from this reviewer.

VERSION 2 – AUTHOR RESPONSE

We would like to kindly thank the reviewer for his recommendation to publish our manuscript and thank the editorial committee for the opportunity to revise our manuscript. We've added the planned systematic search dates and the searched databases to the abstract. At line 31-33 we have clarified that in our study we will exclude qualitative studies. Regarding the PROSPERO number we are unfortunately still waiting to receive our number as the protocol is currently being assessed by their

editorial team. Therefore, we would like to ask your permission to provide the registration number upon reviewing the proof-read? We hope this is not too much of an inconvenience.

Once again, many thanks for the opportunity to publish our protocol.